# Prevalence of skin Neglected Tropical Diseases and superficial fungal infections in two peri-urban schools and one rural community setting in Togo

**Bayaki Saka[1], Panawé Kassang[1], Piham Gnossike[1], Michael G. Head[2]\*, Abla Séfako Akakpo[1], Julienne Noude Teclessou[1], Yvette Moise Elegbede[3], Abas Mouhari-Toure[4], Garba Mahamadou[1], Kokoé Tevi[1], Kafouyema Katsou[1], Koussake Kombaté[1], Stephen L. Walker[5], Palokinam Pitché[1]**

1 Service de dermatologie, CHU de Lomé, Lomé, Togo, 2 Clinical Informatics Research Unit, Faculty of Medicine, University of Southampton, Southampton, United Kingdom, 3 Service dermatologie, CHU de Kara, Kara, Togo, 4 Service de dermatologie, CHR de Kara, Kara, Togo, 5 Faculty of Infectious and Tropical Diseases, London School of Hygiene & Tropical Medicine, London, United Kingdom

\* m.head@soton.ac.uk

**Data Availability Statement:** An anonymized dataset can be downloaded from https://doi.org/10.6084/m9.figshare.21276693.v1.

## Abstract

### Introduction

Skin neglected tropical diseases (NTDs), are endemic and under-diagnosed in many lower-income communities. The objective of this study was to determine the prevalence of skin NTDs and fungal infections in two primary schools and a community setting in rural Togo.

### Method

This was a cross-sectional study that took place between June-October 2021. The two primary schools are located on the outskirts of Lomé, the capital city. The community setting was Ndjéi, in north-east Togo. Study sites were purposively selected. Dermatologists examined the skin of study participants. Diagnosis of skin NTDs were made clinically.

### Results

A total of 1401 individuals were examined, 954 (68.1%) from Ndjéi community, and 447 (31.9%) were children in the schools. Cutaneous skin infections were diagnosed in 438 (31.3%) participants, of whom 355 (81%) were in community settings. There were 105 observed skin NTDs (7.5%). Within the school setting, there were 20 individuals with NTDs (4.5% of 447 participants), and 85 NTDs (8.9%) from 954 community participants. Across all settings 68/1020 (6.7%) NTDs were in children, and 37/381 (9.7%) in adults. In addition, there were 333 observed mycoses (23.8% prevalence). The main cutaneous NTDs diagnosed were scabies (n = 86; 6.1%) and suspected yaws (n = 16, 1.1%). The prevalence of scabies in schools was 4.3%, and 7.0% in the rural community. One case of leprosy was diagnosed in each school and the rural community, and one suspected Buruli Ulcer case in the community. In the school setting, five (6%) children with a skin NTD reported being

**Funding:** The project received a Business Development Grant from the University of Southampton. The funders had no role in study design, data collection and analysis, decision to publish, or preparation of the manuscript.

**Competing interests:** The authors have declared that no competing interests exist.

stigmatised, four of whom had refused to attend school because of their dermatosis. In Ndjéi, 44 (4.6%) individuals reported having experienced stigma and 41 (93.2%) of them missed at least one day of school or work.

## Conclusion

This study shows that the burden of scabies and skin infections such as superficial mycoses is high in the school and rural community settings in Togo, with associated presence of stigma. Improved health promotion and education across institutional and community settings may reduce stigma and encourage early reporting of skin infection cases to a health facility.

## Author summary

This article is a group of conditions called skin Neglected Tropical Diseases (NTDs). The study takes place in Togo, West Africa. There is very little evidence around how widespread NTDs are in Togo. Local dermatologists carried out skin examinations of students in two schools on the edge of Lomé, the capital city. They also examined skin of community residents in rural Togo, in the north-east of the country. We show how the prevalence of fungal skin disease is very high, and also diagnosed numerous cases of scabies (one of the skin NTDs). Cases of leprosy and Buruli Ulcer were also found. Additionally, there were high levels of reported stigma. Our findings show how addressing this burden of disease is vital to improve individual and population health, but also to reduce the socio-economic consequences of these treatable conditions.

## Introduction

Skin Neglected tropical diseases (NTDs) are a group of diseases that are particularly prevalent in many low and middle income countries. According to the World Health Organization (WHO), more than one billion people, mainly in lower-income settings, are affected by one or more of these diseases [1]. Children are more affected than adults, and risk factors include low socio-economic status, crowding, malnutrition, and humidity [2]. Skin NTDs may lead to reduced quality of life and affect psychological wellbeing because of the appearance, functional impairment, discrimination and stigmatisation of the individuals' experience [3]. Residence in rural areas constitutes a population at risk for many cutaneous NTDs, including leprosy [4], Buruli ulcer [5], yaws [6], and scabies [7]. Scabies, caused by the *Sarcoptes scabiei* mite, affects an estimated 455 million people worldwide each year [2].

In many areas of Africa south of the Sahara where skin NTDs are endemic, there are few dermatologists, which reduces the prospects of effective diagnosis of these conditions which are mostly communicable [8]. Educational interventions are known to facilitate early detection of stigmatising or contagious diseases such as leprosy, and to prevent their spread in the community [9]. Community-based studies can also support development of new knowledge around integrated approaches to screening and mass management of certain endemic diseases [10].

The aim of this study was to determine the prevalence of skin NTDs in two schools and one rural community setting, that of village of Ndjéi, in rural north-east Togo. These findings can

be used to provide reliable data that can inform local, national and international decision-making around skin NTD management.

## Methods

### Ethics statement

This study was approved by the Togo Bioethics Committee for Health Research (S2 Supplementary, reference: 012/2021/CBRS of 5 May 2021), and also by the University of Southampton ethics committee (S3 Supplementary, reference ERGO 63498). Written informed consent from participating adults was obtained. For minors, including adolescents, written informed consent were signed by a parent or adult competent guardian after an explanation in a language they understood (French or other local language).

This manuscript is also translated into French (S1 Supplementary). For this cross-sectional study, mobile clinics were set up by dermatologists in Togo, in school and community sites (Fig 1).

The primary schools are based in the Maritime region, located in the urban western and northern areas approximately ten kilometres from the centre of the capital city of Lomé.

The village of Ndjéi, with a population of about 3,000, is located in the Canton of Sirka, in the Kara region, about 400 km north of the capital, Lomé. The nearest large town is Atakpamé, approximately 160km away. Medicines were provided free of charge to participants (see S2 Supplementary). Where necessary, individuals were referred to the appropriate service in the healthcare system. For example, leprosy patients were referred to specific leprosy care centers at the district hospital. Both male and female dermatologists were available across school and community settings.

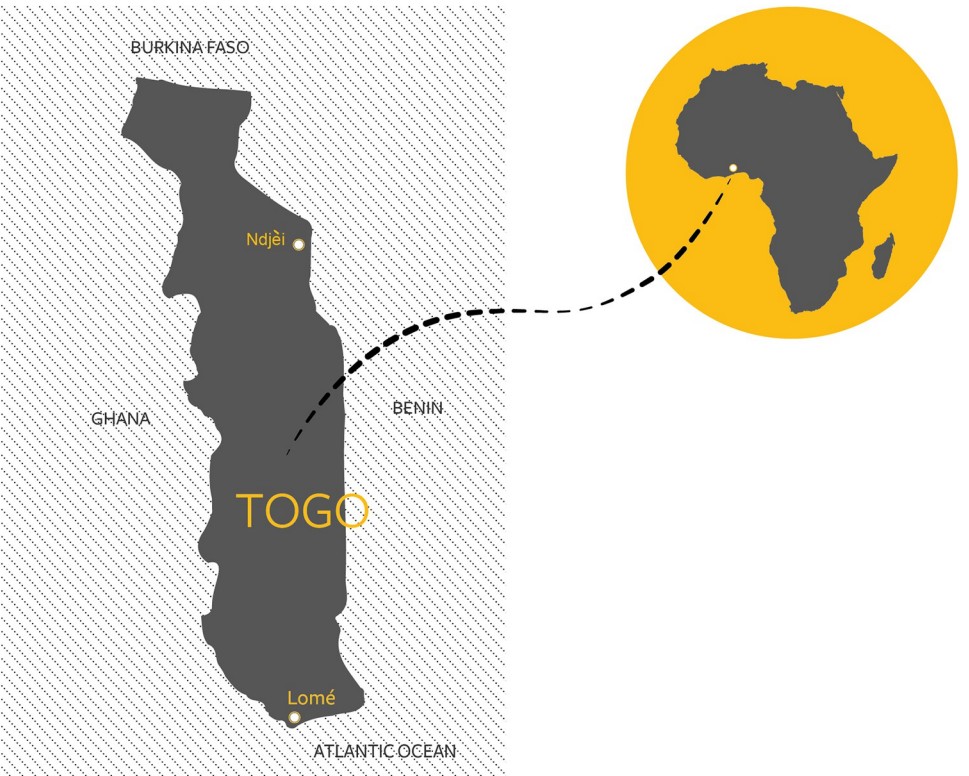

**Fig 1. Location of Lomé, capital city of Togo and where the schools were sited, and the rural community of Ndjei.**
Image drawn within our research group, the Clinical Informatics Research Unit at University of Southampton.

### School environment

The consultations were conducted over three days in June 2021 in the schools of Afiadegnig-ban and Kpédevikopé. The participants were examined in a private room by one dermatologist, with a second examining dermatologist if required (e.g. to confirm a diagnosis). Each of the five dermatologists had a minimum of five years of experience in dermatological medical practice. The skin examination took approximately ten minutes, (with approved Covid 19 precautions). The dermatology team were assisted by two nurses and four community health workers at each site.

### Community environment

The prefectural director of health informed the medical staff of the Ndjéi health centre (nurses, community health workers) that a team of dermatologists would be visiting from 25 to 29 October for mobile consultations. The village chief then informed the population. The population of Ndjéi is estimated at 3,000 people. The study team estimated that they would be able to see approximately one-third of the population during the study. The main activity of the population of Ndjéi is agricultural. A dermatologist attended the primary school to systematically examine all children and their parents/guardians. Another dermatologist carried out similar activity at the secondary school in Ndjéi. The village health centre was used as a consultation site where two dermatologists examined all people who attended for vaccination (national vaccination programme, Covid-19) or any other consultation (malaria, gastro-enteritis).

The study team set up a temporary clinic 3 km from the health centre, in a church. Households were randomly invited to come to the temporary clinic for a skin examination. Household selection was via the random walk method, with guidance from local health service colleagues (for example whether to purposively exclude a household). Of 25 selected households, 19 (76%) agreed to participate in the study. Participants had not previously been examined at other sites. The teams adhered to infection control guidance in relation to COVID-19.

### Sample sizes

Given an estimated 392 and 390 number of students in each school, 3% margin of error, 95% confidence levels, we calculated a required sample size of 287 (3% margin of error, 95% confidence). For the community, given an estimated 3000 population we calculated a required sample size of 788 (3% margin of error, 95% confidence)

### Diagnosis of skin NTDs

The following were considered to be cutaneous NTDs: the dermatoses on the WHO list (https://www.who.int/neglected_diseases/skin-ntds/en/) These were: Buruli ulcer; cutaneous leishmaniasis; post-kala-azar cutaneous leishmaniasis; leprosy; lymphatic filariasis (lymphoedema and hydrocele); mycetoma; onchocerciasis; scabies; yaws; and subcutaneous fungal infections. The study team also recorded the presence of superficial fungal diseases (for example, ringworm). The diagnosis of cutaneous NTDs was exclusively clinical, thus Buruli ulcer and yaws are suspected cases.

### Data collection and analysis

Data were collected using a standardised form for each participant (S3 and S4 Supplementary files). Data collected included socio-demographic information (age, sex, level of education, number of people per household), clinical data (functional signs, type of lesions, site of lesions), diagnosis and questions on stigmatisation. Stigma was noted if the individual

reported experiencing disapproval or prejudice because of their skin problem. The question on the data collection form was "If a skin infection is diagnosed, did the study participant experience stigma as a result of their infection?" (S4 Supplementary). The dermatologist asked if the individual was being treated differently or unfairly as a result of their skin infection, a common approach to gain an indication around levels of stigma [11].

If scabies was diagnosed, further data were collected on the clinical presentation and morphology and site of lesions. The data were entered in EPIDATA software, French version 3.1. Descriptive analyses were performed using Ran version 3.3.2 and results were presented in graphs, tables, frequency and percentages. Quantitative variables were described by means (± standard deviation) and qualitative variables by frequencies and percentages.

## Results

Skin examinations were carried out on a total of 1401 people. This included 954 (68.1%) in the community (Ndjéi). There, 296 (31%) participants were seen at the Ndjéi primary school, 220 (23.1%) at the secondary school, 351 (36.8%) at the village health center and 87 (9.1%) at the temporary clinic. In the Lomé school setting, 447 children (31.9%) were examined out of a total of 782 regularly-enrolled children (and so 57.2% of the school population were participants here).

The mean of the subjects was 10±2.6 years (range 5–17 years) in the school setting and 19 ±15 years (range 1–90 years) in the community setting. The sex ratio (M/F) was 0.9 in both settings. Cutaneous skin infections were diagnosed in 438 (31.3%) patients, of whom 355 (81%) were in community settings (Table 1).

There were 105 observed skin NTDs (7.5%). This included 20 in schools (4.5% of participants in that setting) and 85 (8.9%) in the rural community setting. When considered across all settings by age, there were 68 (6.7%) of NTDs in children, and 37 (9.7%) in adults.

There were 333 observed mycoses (23.8% across all sites). Within schools, there were 63 fungal infections (14.1%), and in the community, 270 fungal infections (28.3%). When considered across all settings by age, there were 259 (25.4%) fungal infections in children, and 74 (19.4%) in adults. There were two cases of leprosy, both being multibacillary (MB). One suspected case of Buruli Ulcer case was diagnosed, with late-stage presentation.

**Table 1. Skin NTDs and mycoses observed, N (%).**

|  | School environment (n = 447) | Community environment (n = 954) | Total (N = 1401) |
|---|---|---|---|
| **Cutaneous NTDs** |  |  |  |
| Yes | 20 (4.5) | 85 (8.9) | 105 (7.5) |
| No | 427 (95.6) | 869 (91.1) | 1296 (92.5) |
| **Type of cutaneous NTDs** |  |  |  |
| Buruli ulcer | 0 (0) | 1 (0.1) | 1 (0.1) |
| Leprosy | 1 (0.2) | 1 (0.1) | 2 (0.1) |
| Scabies | 19 (4.3) | 67 (7.0) | 86 (6.1) |
| Yaw | 0 (0) | 16 (1.7) | 16 (1.1) |
| **Mycoses observed** |  |  |  |
| Total fungal infections | 63 (14.1) | 270 (28.3) | 333 (23.8) |
| *Tinea capitis* | 16 (3.6) | 108 (11.3) | 124 (8.9) |
| *Pityriasis versicolor* | 44 (9.8) | 156 (16.4) | 200 (14.3) |
| *Tinea corporis* | 3 (0.7) | 6 (0.6) | 9 (0.6) |
| **Total skin infections observed** | 83 (18.6) | 355 (37.2) | 438 (31.3) |
| **Participants with no observed skin infections** | 364 (81.4) | 599 (62.8) | 963 (68.7) |

**Table 2. Skin NTDs and mycoses observed, differentiated between children and adults, N (%).**

| | Children (<18 years) (n = 1020) | Adults (≥18 years) (n = 381) | Total (N = 1401) |
|---|---|---|---|
| **Cutaneous NTDs** | | | |
| Yes | 68 (6.7) | 37 (9.7) | 105 (7.5) |
| No | 952 (93.3) | 344 (90.3) | 1296 (92.5) |
| **Type of cutaneous NTDs** | | | |
| Buruli ulcer | 0 (0) | 1 (0.3) | 1 (0.1) |
| Leposy | 1 (0.1) | 1 (0.3) | 2 (0.1) |
| Scabies | 55 (5.4) | 31 (8.1) | 86 (6.1) |
| Yaw | 12 (1.2) | 4 (1) | 16 (1.1) |
| **Mycoses observed** | | | |
| Total fungal infections | 259 (25.4) | 74 (19.4) | 333 (23.8) |
| *Tinea capitis* | 122 (11.9) | 2 (0.5) | 124 (8.9) |
| *Pityriasis versicolor* | 133 (13) | 67 (17.6) | 200 (14.3) |
| *Tinea corporis* | 4 (0.4) | 5 (1.3) | 9 (0.6) |
| **Total skin infections observed** | 327 (32.1) | 111 (29.1) | 438 (31.3) |
| **Participants with no observed skin infections** | 693 (67.9) | 270 (70.9) | 963 (68.7) |

Of these participants, 14 (3.2%) had at least two skin infection at the same time. The most prevalent skin NTD was scabies, observed in 86 patients (6.1%) and suspected yaws in 16 participants (1.1%) (Table 1). The prevalence of scabies was 8.1% in adults compared to 5.4% in children; the prevalence of yaws similar across ages groups (1% in adults compared to 1.2% in children). Superficial fungal infection was more prevalent in children (25.4% in children compared to 19.4% in adults) (Table 2). In the school setting, five (6%) children with cutaneous NTDs reported being stigmatised, and four of them had refused to go to school for one or more days because of this dermatosis. In Ndjei, 44 (4.6%) participants reported having experienced stigma and 41 (93.2%) of them missed at least one day of school or work.

The mean age of scabies patients in the school setting was 10±2 years (range 5 and 13 years) and 20±14 years (range 1 and 72 years) in Ndjei. Across all reported NTDs, pruritus was the main symptom in all participants. Pruritus was more marked at night in 77 participants with scabies (89.5%), and 78 (90.7%) reported concurrent itching in household members. The main lesions noted were papules (77 patients; 89.5%), scratch lesions (74 patients; 86.0%), erosions/ulcerations (39 patients; 45.3%) and scabetic nodules (31 patients; 36%) (Table 3 and S5 Supplementary). These lesions were mainly located on the buttocks (68 cases; 79.1%), wrists (65; 75.6%), and interdigital spaces (56; 65.1%). The lesions were impetiginised in 36 patients and eczematised in 12 others. There were 36 participants (41.9%) who reported being stigmatised because of their scabies infection and 11 of them (30.6%) missed at least one day of school or work because of this stigma.

## Discussion

This study has provided evidence of the prevalence of skin NTDs and fungal infections in school and community-based settings in Togo. There is a very high prevalence of superficial mycoses, with scabies and yaws being the most common skin NTDs.

The prevalence of skin NTDs vary greatly by setting type (for example community, or institution) and by country. In our study, the prevalence of cutaneous NTDs was 7.5%, less than the 17.2% from an Ethiopian hospital study [12]. Many other studies have focused on a single, or small number of, NTDs, yet most of these diseases share some of the same risk factors, can be found concurrently in the same patient, and could arguably be more efficiently investigated

**Table 3. Elementary lesions and lesion sites in scabies patients, N (%).**

| | School environment (n = 19) | Community environment (n = 67) | Total (N = 86) |
|---|---|---|---|
| **Age (years)** | **10±2.6** | **19±14** | - |
| **Sex** | | | |
| Male | 10 (52.6) | 29 (43.3) | 39(45.3) |
| Female | 9 (47.4) | 38 (56.7) | 47 (56.3) |
| **Morphology of lesions** | | | |
| Papules | 11 (57.9) | 66 (98.5) | 77 (89.5) |
| Scratch lesions | 14 (73.7) | 60 (89.6) | 74 (86) |
| Erosions/Ulcerations | 13 (68.4) | 26 (38.8) | 39 (45.3) |
| Scabious nodules | 10 (52.6) | 21 (31.3) | 31 (36) |
| Vesicles/bubbles | 6 (32.6) | 11 (16.4) | 17 (19.8) |
| Pustules | 9 (47.4) | 3 (4.5) | 12 (14) |
| Burrows | 3 (15.8) | 2 (3) | 5 (5.8) |
| Other lesions | 0 (0.0) | 3 (4.5) | 3 (3.5) |
| **Site of lesions** | | | |
| Buttocks | 14 (73.7) | 54 (80.6) | 68 (79.1) |
| Wrists | 13 (68.4) | 52 (77.6) | 65 (75.6) |
| Interdigital spaces | 14 (73.7) | 42 (62.7) | 56 (65.1) |
| Breasts | 6 (31.6) | 27 (40.3) | 33 (38.4) |
| Forearms | 5 (26.3) | 26 (38.8) | 31 (36) |
| Arms | 2 (10.5) | 24 (35.8) | 26 (30.2) |
| Thighs | 0(0) | 21 (31.3) | 21 (24.4) |
| External genitalia | 4 (21.1) | 15 (22.4) | 19 (22.1) |
| Peri-umbilical region | 2 (10.5) | 15 (22.4) | 17 (19.8) |
| Palms of the hands | 0 (0) | 5 (7.5) | 5 (5.8) |
| Feet | 0 (0) | 3(4.5) | 3 (3.5) |
| Other | 4 (21.1) | 9 (13.4) | 13 (15.1) |
| **Complications** | | | |
| Impetiginization | 3 (15.8) | 33 (49.3) | 36 (41.9) |
| Eczematization | 2 (10.5) | 10 (15) | 12 (14) |
| **Stigma** | | | |
| Received stigma? | 3 (15.8) | 33 (49.3) | 36 (41.9) |
| And of those who answered yes to stigma—Missing school or work? | 2 (66.7) | 9 (27.3) | 11 (30.6) |

together [1]. Here, 3.2% of participants had two cutaneous NTDs at the same time. An Ivorian study also reported the frequent association of patients reporting multiple cutaneous conditions, particularly fungal infections alongside one case of leprosy [13]. The recommendations of the WHO NTDs Roadmap indicate that these conditions should be considered alongside each other rather than separately [1].

Fungal diseases (23.8%) were the most prevalent skin infection in our study, followed by scabies (6.1%) and yaws (1.1%). The WHO list of cutaneous NTDs previously referred to 'fungal infections' [14]. However, a more pragmatic definition of skin NTDs typically refers to deep tissue or subcutaneous mycoses, such as mycetoma, rather than also including the superficial fungal diseases like ringworm. A study in Ethiopia found a fungal infection prevalence of 21.1% [12], a level that is similar to our findings. A study in Benin reported a higher prevalence for fungal dermatoses of 49% [15].

The clinical diagnosis of yaws was made in 16 patients in the village of Ndjei, while no cases were found in schools in the urban environment of Lomé. The diagnosis of yaws in this study

was clinical, and its possible that prevalence may be overestimated due to lack of confirmation by diagnostic tests. The skin presentation of primary yaws can be similar to cutaneous leishmaniasis or mycobacterial disease [16]. In a similar study in Côte d'Ivoire, of 15 cases of clinically suspected yaws, the diagnosis was confirmed in only 8 individuals [13].

The prevalence of scabies in schools (4.3%) is similar to the prevalence in studies of schoolchildren in Egypt, Nigeria and Turkey (4.4%, 4.8% and 2.16% respectively) [17–19]. In contrast, a higher prevalence of 39.4% and 17.2% was observed in schoolchildren in India and Cameroon respectively [20,21]. The community prevalence of scabies in this study is higher than in the school setting, but similar to the 6% reported in a Tanzanian rural community [22], and higher than the 2.8% reported in a Cameroonian study [23]. Where community prevalence is high, this increases the risks of scabies outbreaks within the school setting, for example a Ghana study highlighted overcrowded classrooms and sharing of sleeping mats as a likely factor contributing to an outbreak of 92 cases in one school [24].

Pruritus was the main symptom of scabies in our participants, as also reported elsewhere [24,25]. Pruritus, especially nocturnal pruritus and pruritus in families, is typically one of the most suggestive feature of scabies [2]. The buttocks, wrists and interdigital spaces were here observed as the main sites of scabies lesions. These are common sites for a scabies infection to present, though clinical appearance can occur on almost any part of the body, and the presentation may be subtle. This can make diagnosis difficult, with the presentation being under clothing or not reported by the patient, or missed by the healthcare worker. Better knowledge around the most-commonly affected body sites can support simple training methods that aid diagnosis [26].

Bacterial infection (41.9% of cases) was the main complication of scabies in our study. The frequent use of traditional remedies to treat dermatoses could also promote bacterial infection or eczematisation of lesions [27]. Where the pruritus is difficult to conceal, this may be stigmatizing or lead to social exclusion. In an Ethiopian study, almost one third of scabies patients reported stigma [7]. In our study and across all NTDs, 49 people reported having experienced stigma. This stigmatisation and the problem of absenteeism associated with it should encourage the health authorities to make the treatment of cutaneous NTDs free of charge, including consultation fees, and provision of anti-cutaneous NTD drugs. This can help to persuade the population to seek a consultation.

The evidence base around diagnosis, management and identification of skin NTDs is variable, and prevalence data can be inconclusive or out of date. Studies on scabies may more commonly focus on outbreaks or take place in institutional settings [24,28]. This is in part due to higher burdens of disease having been observed, but also due to their relatively simplicity compared with large community surveys. A Liberia study focused on scabies in the community, reporting a prevalence of 9.3% [29], similar to the 7.2% scabies community prevalence observed here.

In 2022, the WHO have revised an NTD roadmap [1], with targets around clear reductions and eliminations of some NTDs by 2030, and a section focused on their epidemiology and prevalence. Much of this global ambition is reliant on donations of medicines such as ivermectin, and for them to be incorporated into Mass Drug Administrations (MDA)s. It is likely that MDAs will have the most impact in higher-burden areas, and thus the data on the community prevalence is vital. In lower burden settings, the WHO report that surveillance and facility-based care may be more appropriate[1].

There are few data on institutional or community prevalence of NTDs in Togo. The country has had significant successes with control programmes, having eliminated dracunculiasis (2011), lymphatic filariasis (2017), and human African trypanosomiasis (2020). The data from this study shows skin NTDs such as scabies and fungal dermatoses are highly-prevalent and

may be harder to eliminate. Scabies may be controlled with mass drug administration of iver-mectin (which is also used to control onchocerciasis). The WHO are developing provisional guidance to around the mass use of ivermectin for scabies control [30].

The main limitations of our study are around the selection of the study population. Within schools, there was reluctance from some parents/guardians to give written consent to allow their children to participate which explains why only 57.2% of the students were examined in the school setting. If children with a skin infection are indeed being stigmatized, then they may have been more likely to be absent at the time of our study, and thus prevalence may be underestimated. Similarly in the rural community, some participant were attending a vaccina-tion clinic at the health centre. Thus, the characteristics of children absent from school and those who did not attend in Ndjei may bias the results. The sample analysed here may not be representative of these populations as a whole. With school and community health programs such as mass drug administrations or vaccine rollouts, uptake is typically higher where there has been extensive community engagement [31]. A sustained programme of health promotion, across school and community settings, may encourage a proactive approach to healthcare-seeking behaviour, plus participation in research studies. Diagnoses were clinical, and no fur-ther diagnostics were performed, thus diagnoses of yaws and Buruli ulcer are suspected rather than confirmed.

## Conclusion

Our study shows that the burden of cutaneous NTDs and fungal infections is high in school and community settings in Togo. Scabies was the most prevalent infection. An integrated management approach as well as mass drug administration programmes may be effective in controlling of these NTDs. However, there should also be consideration around approaches to reduce associated stigma, with improved health promotion and education across institutional and community settings that encourage early reporting to a health facility.

## Supporting information

**S1 Supplementary. French language version of this manuscript.**
(DOCX)

**S2 Supplementary. List of medicines available during the study.**
(DOCX)

**S3 Supplementary. Data collection form used in this study, French version.**
(DOCX)

**S4 Supplementary. Data collection form used in this study, English version.**
(DOCX)

**S5 Supplementary. Further presentation and vizualisation of scabies data.**
(DOCX)

## Acknowledgments

We thank the administrative and health authorities of the Golfe and Binah health districts for allowing us to carry out this study. Thanks to Ashley Heinson, University of Southampton, for his efforts in developing the heat map visualisations of scabies presentations on different body sites. We would like to pay tribute to Dr Mwelecele Ntuli Malecela who died in 2022 and was

Director of the WHO Department of Control of Neglected Tropical Diseases; may she rest in peace.

## Author Contributions

**Conceptualization:** Bayaki Saka, Michael G. Head, Julienne Noude Teclessou, Abas Mouhari-Toure, Garba Mahamadou, Stephen L. Walker, Palokinam Pitché.

**Data curation:** Bayaki Saka, Panawé Kassang, Piham Gnossike, Abla Séfako Akakpo, Julienne Noude Teclessou, Yvette Moise Elegbede, Abas Mouhari-Toure, Garba Mahamadou, Kokoé Tevi, Kafouyema Katsou, Koussake Kombaté.

**Formal analysis:** Bayaki Saka.

**Funding acquisition:** Michael G. Head.

**Methodology:** Bayaki Saka, Panawé Kassang, Michael G. Head, Abla Séfako Akakpo, Garba Mahamadou, Koussake Kombaté, Stephen L. Walker, Palokinam Pitché.

**Project administration:** Piham Gnossike, Michael G. Head, Garba Mahamadou.

**Writing – original draft:** Bayaki Saka, Michael G. Head, Garba Mahamadou, Stephen L. Walker, Palokinam Pitché.

**Writing – review & editing:** Bayaki Saka, Panawé Kassang, Piham Gnossike, Michael G. Head, Abla Séfako Akakpo, Julienne Noude Teclessou, Yvette Moise Elegbede, Abas Mouhari-Toure, Garba Mahamadou, Kokoé Tevi, Kafouyema Katsou, Koussake Kombaté, Stephen L. Walker, Palokinam Pitché.

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
