## [Decision Letter · Decision Letter 0]

25 Sep 2022

Dear Dr Head,

Thank you very much for submitting your manuscript "Prevalence of skin Neglected Tropical Diseases and superficial fungal infections in two peri-urban schools and one rural community setting in Togo" for consideration at PLOS Neglected Tropical Diseases. As with all papers reviewed by the journal, your manuscript was reviewed by members of the editorial board and by several independent reviewers. In light of the reviews (below this email), we would like to invite the resubmission of a significantly-revised version that takes into account the reviewers' comments. 

We cannot make any decision about publication until we have seen the revised manuscript and your response to the reviewers' comments. Your revised manuscript is also likely to be sent to reviewers for further evaluation.

Sincerely,

Paul J. Converse

Academic Editor

Dileepa Ediriweera

Section Editor

Dear Dr. Head,

Your manuscript has been reviewed by three experts in the field and also by myself. The data and results are an interesting summary of convenience samples that may lead to more thorough future studies. Please carefully and fully respond to all of the reviewers' comments to help make the manuscript acceptable for publication. Please also carefully review the text and tables for missing words, and grammatical and spelling errors. Thank you for your submission to PLoS NTD.

Reviewer's Responses to Questions

**Key Review Criteria Required for Acceptance?**

**Methods**

-Are the objectives of the study clearly articulated with a clear testable hypothesis stated?

-Is the study design appropriate to address the stated objectives?

-Is the population clearly described and appropriate for the hypothesis being tested?

-Is the sample size sufficient to ensure adequate power to address the hypothesis being tested?

-Were correct statistical analysis used to support conclusions?

-Are there concerns about ethical or regulatory requirements being met?

Reviewer #1: (No Response)

Reviewer #2: This cannot really be called a prevalence study as almost 40% of the school children invited did not attend because of lack of parental consent and from later figures only 1/3 of the community population was investigated. It is not clear therefore whether attendees were truly respresentative of the whole population 

So without understanding whether those who did not attend had the same rate and distribution of skin disease and Skin NTDs we do not have a true picture of prevalence. The authors briefly discuss this at the end in relation to the school population. 

I note that medicines were provided free of charge – was this for the common skin diseases ? 

“Where necessary, individuals were referred to the appropriate service in the

healthcare system” . Is this in reference to NTDs such as leprosy ?

What were the other skin infections ( briefly) ? 

How was the issue of stigma approached – for instance what questions were asked ? This is critical to understanding the implications of this work and more detail is needed. I presume it is this one from the supplementary material Si «oui» à la question sur la stigmatisation, le participant à l’étude n’a-t-il pas assisté à un ou plusieurs jours de travail ou d’école en raison de la stigmatisation?. There should be a comment in the text – but is this sufficient evidence to indicate stigmatisation ?

Reviewer #3: Hypothesis is clearly stated, and the study design is appropriate to investigate the stated objective. Please clarify the "community setting" (line 22-25) in the introduction. 

Given that more than 50% of the population in the rural setting were school children, it would be good to add subset analysis of school children vs non school children in the rural setting, etc.

Please explain how the choice of schools and the village was random.

No concerns about ethical or regulatory requirements.

**Results**

-Does the analysis presented match the analysis plan?

-Are the results clearly and completely presented?

-Are the figures (Tables, Images) of sufficient quality for clarity?

Reviewer #1: (No Response)

Reviewer #2: Page 14 scabies furrows - ? Burrows.

This sentence is a bit confusing – “These fungal diseases are not always considered as neglected

 diseases, but they have previously been referred to in the WHO list of cutaneous NTDs”.

Superficial fungal infections are not listed by WHO as Skin NTDS. The issue here is that if skin examination is used as an entry point for NTD detection, for instance as in the new Framework programme published this year, it is a pragmatic and practical approach to include the common prevalent skin diseases such as superficial mycoses when dispensing treatment rather than to ignore them. So they are included in, for instance, the WHO App and training manual but are not listed as NTDs

Reviewer #3: The analysis of the data, especially the given percentages are not clear and partly faulty. Please correct. State clearly what the denominator is.

Line 34 the percentages in parentheses behind the numbers are incorrect instead of 6.7% it should for example be 4.4%

In the text it is described, that 14 patients had more than 1 skin infection, this is not reflected in table 1 and table 2 the number of skin infections plus the number of patients without a skin infections adds up to 100%, if there were 14 patients with more than 1 skin infection, there would need to be at least 14 skin infections more, or the number of patients without a skin infection would need to be higher.

Table 3 last row, the percentages don't fit. Please recheck all percentages and make clear the percentage of what (sub)population it is

**Conclusions**

-Are the conclusions supported by the data presented?

-Are the limitations of analysis clearly described?

-Do the authors discuss how these data can be helpful to advance our understanding of the topic under study?

-Is public health relevance addressed?

Reviewer #1: (No Response)

Reviewer #2: In discussing the accuracy of the diagnosis of yaws can you provide an example of a disease that might be confused with yaws ?

The authors point out the main limitation of this study which is the comparatively low participation rate in schools . Without examining non attenders it is not possible to comment on the accuracy of the prevalence figures although these may not have been affected. But it would be helpful to explore and suggest ways of increasing participation for the future as this is critical information for the success of Skin NTD programmes.

Reviewer #3: Please add in the discussion a part discussing other possibilities/more in depth how the low percentage of school children could have been effected by the fact, that people with skin NTD rather stay at home, among other theories maybe.

Add, that in the rural setting a subset was investigated, because these were people, who were already coming to the health center, this is a subset of the population, which could differ from the overall population.

Line 258 - add percentage of people reported stigma/all people with NTDs

Line 287 - please reiterate further on the limitations of the study, e.g. on the small sample size

**Editorial and Data Presentation Modifications?**

Reviewer #1: (No Response)

Reviewer #2: (No Response)

Reviewer #3: Consider using a map, which shows also other cities in Togo, to show where the remote community (Ndjéi) is.

Line 69 - add how climate and rainfall are risk factors and what climate is a risk factor.

**Summary and General Comments**

Reviewer #1: Line 77 - not all of the skin NTDs are communicable (for example BU is likely from the environment as are most of the fungal diseases) and it is not clear that it is the absence of dermatologists that creates a barrier to their control. 

Line 79- the reference appears to be about leprosy; I am not aware of any good evidence that increased access to dermatologists (which is undoubtedly a good thing) helps break chains of transmission of yaws or scabies. Please reword.

Methods - were dermatologists of both genders available? 

- What if any attempts were made to standardise the performance of the dermatologists and the terminology they used?

How was the random selection of households done? Simple random sampling? Random walk? Where was the randomisation list generated from.

It is not possible to make a clinical diagnosis of yaws reliably - this has been shown in numerous papers. Similarly BU can not be diagnosed reliably solely on clinical grounds. I would suggest you reword the methods to make clear that a suspected diagnosis could be made but not an actual one. Tables and references to these diagnoses in the results and discussion also need to be amended. 

- In addition for yaws it is critical to differentiate between papillomas (specific, likely to be yaws) and ulcers (non specific, most commonly wont be yaws)

How were the questions on stigma asked/collected? Is this a validated tool.

The information on consent is provided twice in duplicate (line 101 and then again in the ethics section) please remove one of these.

Results - what is a furrow - I presume this is a typo and is meant to be burrow.

Discussion - I do not think you should be referring to a diagnosis of yaws here and certianly not claiming it was common without a) some details on papillomas vs ulcers and b) given the complete absence of diagnostic testing

Line 225 I am not aware that WHO has ever regarded superficial mycoses as NTDs and the reference you provide for this statmeent (13) does not suggest this either. Please remove/reword this. 

The discussion is quite long (>1200 words) and I would suggest benefit from editing to make it more concise. 

Where are data available from - please provide this informaiton

Reviewer #2: While I recognise the difficulty of carrying out studies of this nature some attempt should be made to show that the population examined were representative of the whole

Reviewer #3: Good study overall, which gives insight into an understudied topic and adds value to the community, adding important information regarding the prevalence of skin NTDs in another african country. The study is limited to a low number of participants and the methods used are subjective. Not sufficient information about the training of the doctors is given, and as clinical diagnoses were the only method to diagnose a high false positive/negative rate must be assumed. The relevance of the sub-analysis is questionable as no major conclusions were found. Research and publication ethics seem to have been respected. A good study requiring some more work on the analysis part.

Data for the socio-demographic information collected lacks.

PLOS authors have the option to publish the peer review history of their article (what does this mean?). If published, this will include your full peer review and any attached files.

Reviewer #1: No

Reviewer #2: No

Reviewer #3: No
---

## [Decision Letter · Decision Letter 1]

7 Dec 2022

Dear Dr Head,

We are pleased to inform you that your manuscript 'Prevalence of skin Neglected Tropical Diseases and superficial fungal infections in two peri-urban schools and one rural community setting in Togo' has been provisionally accepted for publication in PLOS Neglected Tropical Diseases.

Best regards,

Paul J. Converse

Academic Editor

Dileepa Ediriweera

Section Editor

Dear Dr. Head and colleagues,

Congratulations on the acceptance of your manuscript for publication in PLoS NTD. At your discretion, you may wish to attend to the comments of Reviewer 3 before the paper goes to press. The suggestions should help clarify your message for all readers.

Reviewer's Responses to Questions

**Key Review Criteria Required for Acceptance?**

**Methods**

-Are the objectives of the study clearly articulated with a clear testable hypothesis stated?

-Is the study design appropriate to address the stated objectives?

-Is the population clearly described and appropriate for the hypothesis being tested?

-Is the sample size sufficient to ensure adequate power to address the hypothesis being tested?

-Were correct statistical analysis used to support conclusions?

-Are there concerns about ethical or regulatory requirements being met?

Reviewer #1: (No Response)

Reviewer #2: The authors have addressed my concerns

Reviewer #3: Please explained the paragraph "Sample Size", what is meant with required in this context? Required for what purpose?

**Results**

-Does the analysis presented match the analysis plan?

-Are the results clearly and completely presented?

-Are the figures (Tables, Images) of sufficient quality for clarity?

Reviewer #1: (No Response)

Reviewer #2: The authors have addressed my concerns

Reviewer #3: Tables are better now.

The results part would profit from having the percentages explained in the text instead of behind the numbers. E.g. line 185 instead of "There were 105 observed skin NTDs (7.5%)." Write "In 7.5% of the examined study population NTDs were observed" etc.

**Conclusions**

-Are the conclusions supported by the data presented?

-Are the limitations of analysis clearly described?

-Do the authors discuss how these data can be helpful to advance our understanding of the topic under study?

-Is public health relevance addressed?

Reviewer #1: (No Response)

Reviewer #2: The authors have addressed my concerns

Reviewer #3: (No Response)

**Editorial and Data Presentation Modifications?**

Reviewer #1: (No Response)

Reviewer #2: The authors have addressed my concerns

Reviewer #3: (No Response)

**Summary and General Comments**

Reviewer #1: I am satisifed appropriate changes have been made

Reviewer #2: The authors have addressed my concerns

Reviewer #3: The quality and usability of the provided data set could be improved if the abbreviations used in the data set would be explained.

PLOS authors have the option to publish the peer review history of their article (what does this mean?). If published, this will include your full peer review and any attached files.

Reviewer #1: No

Reviewer #2: No

Reviewer #3: **Yes: **Oliver Komm

---

## [Editor Report · Acceptance letter]

14 Dec 2022

Dear Dr Head,

We are delighted to inform you that your manuscript, "Prevalence of skin Neglected Tropical Diseases and superficial fungal infections in two peri-urban schools and one rural community setting in Togo," has been formally accepted for publication in PLOS Neglected Tropical Diseases.

Best regards,

Shaden Kamhawi

co-Editor-in-Chief

Paul Brindley

co-Editor-in-Chief
